# EchoQA: A Large Collection of Instruction Tuning Data for Echocardiogram Reports

**Lama Moukheiber***
Massachusetts Institute of Technology

**Mira Moukheiber***
Massachusetts Institute of Technology

**Dana Moukheiber**
Massachusetts Institute of Technology

**Jae-Woo Ju**
Seoul National University

**Hyung-Chul Lee**
Seoul National University

## Abstract

We introduce a novel question-answering (QA) dataset using echocardiogram reports sourced from the Medical Information Mart for Intensive Care database. This dataset is specifically designed to enhance QA systems in cardiology, consisting of 771,244 QA pairs addressing a wide array of cardiac abnormalities and their severity. We compare large language models (LLMs), including open-source and biomedical-specific models for zero-shot evaluation, and closed-source models for zero-shot and three-shot evaluation. Our results show that fine-tuning LLMs improves performance across various QA metrics, validating the value of our dataset. Clinicians also qualitatively evaluate the best-performing model to assess the LLM responses for correctness. Further, we conduct fine-grained fairness audits to assess the bias-performance trade-off of LLMs across various social determinants of health. Our objective is to propel the field forward by establishing a benchmark for LLM AI agents aimed at supporting clinicians with cardiac differential diagnoses, thereby reducing the documentation burden that contributes to clinician burnout and enabling healthcare professionals to focus more on patient care.

## 1 Introduction

Echocardiography is the most prevalent noninvasive technique for assessing heart function and detecting heart diseases. It plays a critical role in clinical cardiology, consistently guiding decision-making processes [1]. Echocardiography is essential for diagnosing diseases, stratifying risks, and evaluating treatment efficacy. The diagnostic reports generated from these tests provide rich clinical data, vital for diagnosing and managing various cardiac conditions [2]. The growing demand for diagnostic echocardiograms makes it difficult to manage and interpret the increasing volume of data, and utilizing AI-powered algorithms can reduce clinician workload.

The advent of large language models (LLMs) holds the potential to transform the field of cardiology. LLMs have been utilized across various natural language processing tasks, such as question-answering (QA), text summarization, and language translation, often in zero-shot and few-shot scenarios [3, 4]. In-context learning (ICL) enables the models to tackle new tasks with only a few task demonstrations, like in three-shot prompting, without the need to update model parameters [4]. Moreover, transforming tasks related to understanding and generating natural language into clear instructions enhances the

---

*Equal contribution.

SafeGenAI Workshop @ 38th Conference on Neural Information Processing Systems (NeurIPS 2024).

ability of LLMs to follow domain-specific directives and improve their performance on downstream tasks[5, 6]. Open-source models like Llama [7] and Mistral [8] have demonstrated significant potential in this area.

There is a gap in developing large language models (LLMs) that are trained and evaluated on real-world medical data, such as echocardiogram reports with ground-truth answers, which stems from the reliance on synthetic data or data from medical licensing exams [9, 10]. This limitation has hindered progress of AI in the cardiology space. However, with the recent advancements in instruction-tuning capabilities of LLMs, there is now an opportunity to leverage real-world clinical datasets to create more accurate and context-aware models, addressing the specific needs of cardiologists in their diagnostic workflows[11, 12]. Tasks that cardiologists perform while interacting with patients—such as generating differential diagnoses on their computers from various clinical sources like laboratory results or echocardiographic imaging data—can now be streamlined, reducing the burden of documentation[13–17], improving clinician job satisfaction[18] and allowing clinicians to focus more on patient care[19–21].

Furthermore, addressing algorithmic bias is crucial in healthcare before model deployment. Most studies have incorporated protective attributes, such as race, gender, and age, for fairness auditing of healthcare algorithms [22, 23]. Beyond these common attributes, analyzing social determinants of health could assist in mitigating disparities in patient care[24–26]. Furthermore, incorporating social determinants into fairness audits could help assist with regulations like Section 1557 of the Affordable Care Act, which mandates that healthcare providers and payers ensure their algorithms do not discriminate [27]. Moreover, social determinants of health can help clinicians provide more individualized diagnoses in cardiac care by considering the broader context of patients' living conditions and lifestyle factors.

Based on the challenges aforementioned, our work makes the following three contributions:

- *Development of EchoQA:* We present EchoQA, the largest open-access, real-world patient question-answering dataset for echocardiography, meticulously developed by expert clinicians. Our aim is to propel the medical field by creating a foundation for training LLM-based AI agents that will assist cardiologists in their daily workflows. EchoQA also provides researchers and practitioners with the opportunity to test and compare different machine learning approaches for differential diagnosis.

- *Zero-shot, Few-shot and Instruction Fine-Tuning Evaluations:* Leveraging the EchoQA dataset, we validate its utility by fine-tuning a variety of LLMs, encompassing both general-purpose and medical-domain models, and comparing their performance to zero-shot setups. Additionally, for comparison we conduct zero-shot and three-shot evaluations on commercial LLMs. Furthermore, we release the best-performing echocardiogram model, *Echo-Mistral*, making it accessible to the wider research community.

- *Fairness Audits on Social Determinants of Health:* To investigate algorithmic bias, we use social determinants of health to enable more fine-grained audits of algorithmic fairness. These evaluations provide critical insights into potential disparities often overlooked in LLM studies, promoting health equity.

## 2   Related Work

**Medical question answering datasets.** Medical question-answering benchmark datasets have been developed to address different aspects of medical information retrieval and understanding. Examples include datasets designed for medical licensing exams and conceptual medical knowledge, such as MedQA, JAMA Clinical Challenge, MedBullet, and MMLU Clinical Topics [9, 28, 29]. Additionally, literature-based QA datasets, such as PubMedQA, consist of biomedical research questions derived from PubMed abstracts [30]. On the other hand, datasets like HealthSearchQA, LiveQA, and MedicationQA provide insights into medical information needs from a consumer perspective [31–33]. More specifically, MedicationQA addresses questions related to medications and their uses, aiding in pharmaceutical information retrieval [33]. QA datasets utilizing real-world medical data from electronic health records include emrQA, which consists of factual questions with answers derived from discharge summary reports in the i2b2 dataset [34]. Similarly, RadQA consists if radiology-related questions commonly encountered in clinical practice, using data extracted from radiology reports in the MIMIC database [35]. However, none of these datasets include questions

a echocardiogram reports, which differ significantly in semantic content and vocabulary. Table 2 provides a summary of the medical QA datasets described above.

Table 1: Overview of medical Question-Answering (QA) datasets by domain, QA type, and use of real-world data. QA types include Multiple Choice (predefined answers) and Long-form (free-text responses).

| Dataset | Domain | QA Type | Real-world medical data |
|---|---|---|---|
| MedQA [9] | Medical Board Exams (USMLE) | Multiple choice | |
| JAMA Clinical Challenge [29] | Exam for clinical cases (JAMACC) | Multiple choice | |
| MedBullet [29] | Medical Board Exams (USMLE) | Multiple choice | |
| MMLU Clinical Topics [9] | Medicine and biology-related topics | Multiple choice | |
| PubMedQA [30] | Literature-based (PubMed abstracts) | Multiple choice | |
| HealthSearchQA [31] | Consumer searched questions | Long-form | ✓ |
| LiveQA [32] | Consumer health | Long-form | ✓ |
| MedicationQA [33] | Consumer questions about medications | Long-form | ✓ |
| emrQA [34] | Discharge reports (i2b2 data) | Long-form | ✓ |
| RadQA [35] | Radiology reports (MIMIC data) | Long-form | ✓ |
| **EchoQA (Ours)** | Echocardiography reports (MIMIC data) | Long-form | ✓ |

**LLM and echocardiography.** There is limited research on the use of large language models (LLMs) specifically within cardiology. One work introduced EchoGPT, a fine-tuned Llama-2 model [7] employing Quantized Low-Rank Adaptation (QLoRA) to assist with echocardiography report summarization and initial drafting of reports for clinician review, effectively streamlining the reporting workflow [36]. Further, prior studies indicate that general-purpose LLMs, such as ChatGPT, struggle with echocardiography board review questions, highlighting the need for specialized training to enhance performance in cardiology applications [37]. However, these efforts do not establish a framework for assisting clinicians in the differential diagnosis of cardiac abnormalities.

**Fairness audits.** While progress has been made in addressing algorithmic fairness in healthcare, most studies have focused primarily on biases related to protected attributes such as age, gender, and race [22, 38]. Recent research emphasizes the need to examine biases from multidimensional perspectives, evaluating fairness through the intersectionality of social determinants and social identities to provide a deeper understanding beyond the socially constructed nature of attributes like race and gender [26, 39]. Studies have also incorporated social determinants of health offering insights into the processes driving disparities in machine learning models [25, 40]. We leverage social determinants of health to conduct fine-grained audits of algorithmic fairness on general, biomedical, and closed-source LLMs for cardiac diagnostic support. With that, we hope to account for the broader context of individuals' lives focusing on the conditions in which people are born, grow, live, work, and age, as well as the broader social, economic, and environmental factors that influence health, ultimately assisting clinicians in making more informed and personalized decisions in cardiac diagnosis for individual patients.

## 3 Experimental Setup

### 3.1 Dataset

We curate a question-answering dataset sourced from the Medical Information Mart for Intensive Care (MIMIC-IV) database, a de-identified clinical dataset comprising over 80,000 echocardiogram reports collected at Beth Israel Deaconess Medical Center between 2012-2019 [41], providing a rich resource that can support differential diagnosis and enhance diagnostic decision-making for cardiac abnormalities.

The echocardiogram reports include details on specific heart structures, such as the left atrium, right atrium/interatrial septum, left ventricle, right ventricle, mitral valve, aortic valve, and tricuspid valve. Each patient's echocardiography report is processed to extract unique sentences for each heart structure. Following [42], clinical experts identify diverse abnormalities described in the sentences extracted for each heart structure, and assign levels ranging from -3 to 3 for each identified abnormality. These levels are based on standardized diagnostic criteria established by the American Society of Echocardiography[43–45] , indicating both the category and severity level of the abnormality. A category of -3 indicates that the study is inadequate for evaluating the cardiac abnormality. A category

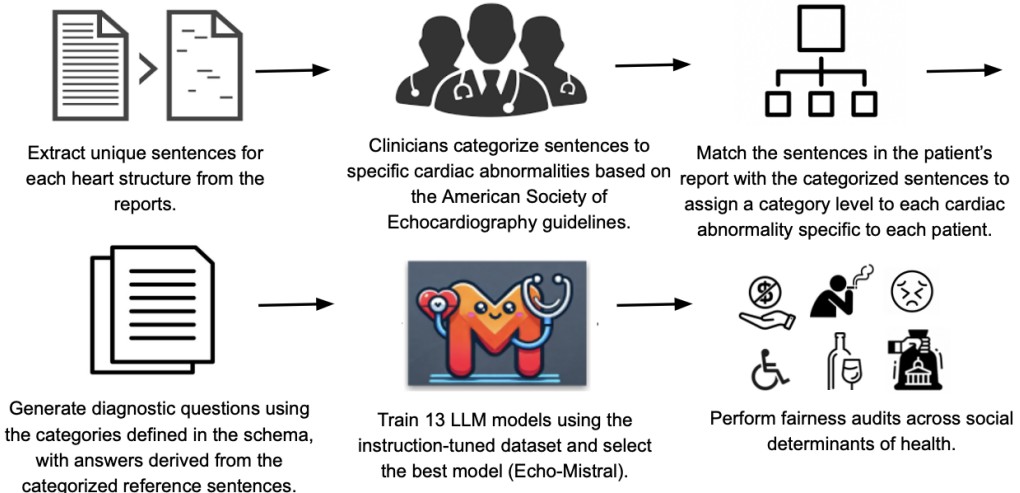

Figure 1: Workflow of the methodology.

of 0 is used when the study is adequate for evaluating cardiac function but reveals no abnormalities. Sentences describing abnormal function without specifying severity are assigned a category of -2. Severity levels are categorized as 1 for mild, 2 for moderate, and 3 for severe. For specific features, such as left ventricular cavity size, left ventricular systolic function, and right ventricular cavity size, a category of -1 indicates hyperdynamic left ventricular systolic function or a small cavity size for the left or right ventricle.

The sentences in the patient's notes are then matched with the sentences categorized for each abnormality to enable the assignment of an abnormality category level for each patient. When multiple sentences for the same abnormality in the patient's notes match different severity levels—mild, moderate, or severe—the highest category level is retained to prioritize the most clinically significant finding. In cases where conflicting category levels derived from sentences in the patient's notes are identified for the same abnormality, a placeholder value of -50 is assigned to indicate ambiguity or disagreement in the abnormality categorization. As illustrated in Figure 2, using these categories, diagnostic questions, such as "Is the study adequate to assess left ventricular systolic dysfunction?" are generated. The answers to these questions are derived directly from the sentences categorized for each cardiac abnormality, resulting in more than 700,000 question-answer pairs, with the categories depicted in Table 2.

This data curation incorporates clinical expertise to establish relevant cardiac diagnostic questions and build cardiac abnormality categorizations from patients' echocardiogram notes, while addressing potential errors in medical documentation to ensure accurate answers for individualized cardiac differential diagnosis. Hence, it establishes a gold standard, enhancing the instruction-following capabilities of large language models in the differential diagnosis of cardiac abnormalities, supporting clinical decision-making, alleviating clinician burnout from documentation, and enabling more physician-patient interaction. The question-answering dataset will be hosted on PhysioNet, an NIH-funded health data repository [46]. Figure 1 illustrates the curation, validation, and auditing process of the instruction-tuned dataset.

## 3.2  Model Inference & Training

To validate the value of the training data, we employ supervised fine-tuning (SFT) on a diverse selection of recent open-source and biomedical domain-specific large language models (LLMs) and compare their performance against zero-shot setups. Additionally, we evaluate closed-source models in zero-shot and three-shot setups, exploring the potential for three-shot configurations to sustainably improve the performance of closed-source LLMs. For open-source general models, we utilize Llama-3-8B [7], Mistral-7B [8], Phi-3-mini [47], Zephyr-7B [48], and Falcon-7B [49]. In the biomedical domain, we leverage specialized open-source models such as BioMistral-7B[50],

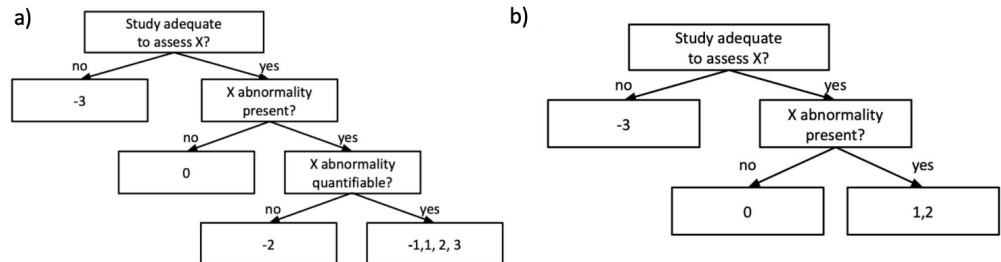

Figure 2: Categorization of cardiac abnormalities. X represents a specific cardiac abnormality. a) The schema includes the following cardiac abnormalities: right atrial pressure; tricuspid valve regurgitation, tricuspid valve stenosis, and pulmonary hypertension; right ventricular systolic function, right ventricular cavity, and right ventricular wall; left atrial cavity; mitral valve regurgitation and mitral valve stenosis; left ventricular systolic function, left ventricular cavity, left ventricular wall, left ventricular diastolic function, left ventricular outflow tract obstruction, and left regional wall motion abnormality; and aortic valve regurgitation and aortic valve stenosis. b) The schema includes other right ventricular and atrial abnormalities: right ventricular pressure overload and right ventricular volume overload; and right atrial enlargement.

Table 2: Cardiac abnormalities found in the echocardiogram reports.

| Cardiac Abnormalities | Number of QA's |
|---|---|
| **Right atrial abnormalities** | |
| Right atrial enlargement | 45,254 |
| Right atrial pressure | 2,371 |
| **Tricuspid valve abnormalities** | |
| Tricuspid valve regurgitation | 13,332 |
| Tricuspid valve stenosis | 19,509 |
| Pulmonary hypertension | 21,376 |
| **Right ventricular abnormalities** | |
| Right ventricular systolic function | 74,236 |
| Right ventricular cavity | 71,971 |
| Right ventricular volume overload | 5,075 |
| Right ventricular pressure overload | 5,065 |
| Right ventricular wall | 7,316 |
| **Left atrial abnormalities** | |
| Left atrium cavity | 14,425 |
| **Mitral valve abnormalities** | |
| Mitral valve stenosis | 38,044 |
| Mitral valve regurgitation | 53,205 |
| **Left ventricular abnormalities** | |
| Left ventricular systolic function | 64,305 |
| Left ventricular cavity | 64,354 |
| Left ventricular wall | 64,295 |
| Left ventricular diastolic function | 5,769 |
| Left ventricular outflow tract obstruction | 40,697 |
| Left regional wall motion abnormality | 39,310 |
| **Aortic valve abnormalities** | |
| Aortic valve stenosis | 61,451 |
| Aortic valve regurgitation | 59,884 |
| **Total** | **771,244** |

M42-health [51], PMC-LLaMa-13B [52], and Meditron-7B [53], which are designed to understand biomedical terminology and context derived from medical abstracts and texts. Additionally, we include proprietary models, such as Amazon-titan [54], Claude [55], Cohere [56], and GPT-4o [57],

to provide a comprehensive comparison across different model types and domains. The closed-source models are deployed on Azure OpenAI [58] and Amazon Bedrock [59]to ensure HIPAA compliance.

Due to computational limitations, we sample 10,000 subjects and divide the data into a 70% training set, a 10% validation set, and a 20% testing set, ensuring that the data for each subject is contained in only one set. The training set is used for fine-tuning the model, while the testing set is used for inference. Both training and inference datasets are processed and tokenized, and the model is configured with dynamic token embedding resizing to accommodate task-specific tokens effectively. The prompts used for zero-shot and three-shot questions answering are shown in Figure 6 and Figure 7 in the Appendix.

For supervised fine-tuning, we train the models for one epoch with a learning rate of 2e-4, using a cosine learning rate schedule with a 1% warm-up ratio to stabilize the initial training phase. Training is performed using the AdamW optimizer, utilizing gradient accumulation steps of 4 to simulate larger batch sizes and optimize memory usage. To enhance training efficiency, we employ Low-Rank Adaptation (LoRA) for parameter-efficient fine-tuning. This includes setting the scaling factor (LoRA alpha) to 16 to control the influence of task-specific adaptations, applying a 10% dropout rate to prevent overfitting, and using a rank of 64 to enable task-specific adaptation with minimal additional parameters. Additionally, we employ the BitsAndBytes quantization technique with Normal Float 4 (NF4) for numerical stability and Brain Floating Point 16-bit (bfloat16) for faster computations. Fine-tuning is conducted on NVIDIA A100 80GB GPUs, utilizing model sharding to efficiently distribute computational resources.

## 3.3 Automated Model Evaluation of LLM Responses

We conduct a comprehensive analysis of our model's performance using quantitative metrics. We employ BLEU score, to measure the precision of n-grams between the generated and reference answers [60]. To assess the balance between precision and recall, we utilize the average F1 Score [61]. We apply the ROUGE-1 and ROUGE-2 metrics to evaluate the overlap of unigrams and bigrams, respectively, between the generated and reference answers, thereby assessing lexical similarity across different levels of granularity[62]. Additionally, we use the ROUGE-L metric to measure the longest common subsequence, indicating the extent to which the generated answer aligns with the reference in terms of in terms of sequential structure and lexical overlap[63]. Lastly, we utilize the average METEOR Score, which evaluates precision and recall while incorporating linguistic features such as synonyms and stemming[64].

## 3.4 Clinician Evaluation of LLM Responses

We identify the best model for differential diagnosis as Mistral-7b (fine-tuned). Clinicians evaluate correctness to determine whether a response is correct or incorrect in aiding differential diagnosis. In total, 1,500 notes, along with query and answer pairs, are reviewed. Of these, 1,485 responses are deemed correct, while the remaining responses are deemed incorrect. Responses are classified as incorrect under certain conditions. First, when the generated response includes a different abnormality than the one addressed in the question. For example, if the question asks about right ventricular pressure overload but the answer discusses pulmonary hypertension. Second, a response is deemed incorrect if it includes irrelevant information that does not assist with the diagnosis. For instance, if the generated answer includes "no mass on tricuspid valve" instead of describing a "normal tricuspid valve leaflet," which is more relevant to diagnosing tricuspid valve regurgitation. Another type of incorrect response occurs when the generated answer fails to prioritize the highest severity level for a specific abnormality. For example, the left ventricular wall is incorrectly classified as normal instead of mild when the note contains multiple sentences describing varying severity levels. Finally, a response is considered incorrect if it includes the correct diagnosis but also adds unrelated diagnoses, compromising the clarity and quality of the answer.

## 3.5 Fairness Audits

We perform fairness audits by examining social health attributes, as these factors provide insights into the conditions in which individuals live —critical influences on a person's health and well-being. To perform these audits, we utilize census tract-level SDOH data from the MIMIC dataset [65]. Our analysis investigates fairness disparities across subgroups defined by societal attributes, such as

whether a patient lives in areas with high unemployment rates, relies heavily on public assistance or food stamps, includes adults who are heavy drinkers or smokers, or reports experiencing mental distress or having a disability. We discretize the dimensions into high, upper middle, lower middle, low groups, based on the quantile of the distribution for each dimension. We select the best-performing model to evaluate bias. For each LLM, bias across various dimensions is assessed using the F1 equality difference metric in [66], which measures the average absolute difference between the f1 of individual social groups and the overall f1 across all groups within the corresponding social category. In particular, for a dimension $D$ and its associated set of demographic groups $\mathcal{G}^D = \{\mathcal{G}_1^D, \mathcal{G}_2^D, \ldots\}$, F1 equality difference $= \frac{1}{|\mathcal{G}^D|} \sum_{\mathcal{G}_i^D \in \mathcal{G}^D} |F1(\mathcal{G}_i^D) - F1(\mathcal{G}^D)|$.

# 4   Results & Discussion

Table 3: Performance metrics for open-source biomedical models, open-source general models, and closed-source general models averaged across 3 runs (higher scores with tighter error margins are better). Fine-tuned open source models are compared to their baseline zero-shot models. Closed-source models are compared to 0-shot and 3-shot in-context learning models. Bolded numbers depict the best model in each of the categories including the open-source biomedical and general models, as well as closed-source models.

| Evaluation Metric | BLEU | ROUGE-1 | ROUGE-2 | ROUGE-L | F1 | METEOR |
|---|---|---|---|---|---|---|
| **Open-source (biomedical)** | | | | | | |
| BioMistral-7B (0-shot) | $0.03957 \pm 0.00000$ | $0.24143 \pm 0.00000$ | $0.12638 \pm 0.00000$ | $0.22213 \pm 0.00000$ | $0.25368 \pm 0.00000$ | $0.32930 \pm 0.00000$ |
| BioMistral-7B (fine-tuned) | $\mathbf{0.68290 \pm 0.00446}$ | $\mathbf{0.96694 \pm 0.00165}$ | $\mathbf{0.96100 \pm 0.00222}$ | $\mathbf{0.96659 \pm 0.00174}$ | $\mathbf{0.96699 \pm 0.00168}$ | $\mathbf{0.95483 \pm 0.00188}$ |
| M42-health (0-shot) | $0.18000 \pm 0.00110$ | $0.60429 \pm 0.00071$ | $0.50154 \pm 0.00137$ | $0.57618 \pm 0.00044$ | $0.60747 \pm 0.00068$ | $0.64191 \pm 0.00060$ |
| M42-health (fine-tuned) | $0.31223 \pm 0.03710$ | $0.71158 \pm 0.07130$ | $0.64302 \pm 0.09202$ | $0.69965 \pm 0.07748$ | $0.71220 \pm 0.07127$ | $0.70233 \pm 0.07844$ |
| Meditron7B (0-shot) | $0.00023 \pm 0.00006$ | $0.08442 \pm 0.00244$ | $0.02637 \pm 0.00084$ | $0.07909 \pm 0.00216$ | $0.08637 \pm 0.00269$ | $0.06323 \pm 0.00235$ |
| Meditron7B (fine-tuned) | $0.00104 \pm 0.00036$ | $0.11387 \pm 0.02270$ | $0.02735 \pm 0.00714$ | $0.10594 \pm 0.02103$ | $0.11666 \pm 0.02126$ | $0.08680 \pm 0.01295$ |
| PMC-llama-13B (0-shot) | $0.00287 \pm 0.00033$ | $0.08877 \pm 0.00133$ | $0.01519 \pm 0.00052$ | $0.08428 \pm 0.00147$ | $0.09030 \pm 0.00140$ | $0.06801 \pm 0.00078$ |
| PMC-llama-13B (fine-tuned) | $0.00478 \pm 0.00153$ | $0.09885 \pm 0.01602$ | $0.02617 \pm 0.01106$ | $0.09409 \pm 0.01377$ | $0.10138 \pm 0.01725$ | $0.08549 \pm 0.02312$ |
| **Open-source (general)** | | | | | | |
| Llama-8B (0-shot) | $0.07216 \pm 0.00139$ | $0.27047 \pm 0.00141$ | $0.21111 \pm 0.00144$ | $0.26227 \pm 0.00117$ | $0.29686 \pm 0.00145$ | $0.48359 \pm 0.00193$ |
| Llama-8B (fine-tuned) | $0.62419 \pm 0.02431$ | $0.95415 \pm 0.00374$ | $0.94527 \pm 0.00499$ | $0.95408 \pm 0.00381$ | $0.95419 \pm 0.00373$ | $0.93504 \pm 0.00543$ |
| Mistral-7B (0-shot) | $0.06410 \pm 0.00000$ | $0.30277 \pm 0.00000$ | $0.19364 \pm 0.00000$ | $0.28000 \pm 0.00000$ | $0.31844 \pm 0.00000$ | $0.46702 \pm 0.00000$ |
| Mistral-7B (fine-tuned) | $0.70062 \pm 0.00185$ | $\mathbf{0.97942 \pm 0.00042}$ | $\mathbf{0.97464 \pm 0.00054}$ | $\mathbf{0.97913 \pm 0.00038}$ | $\mathbf{0.97944 \pm 0.00042}$ | $\mathbf{0.96519 \pm 0.00048}$ |
| Phi-mini (0-shot) | $0.03595 \pm 0.00000$ | $0.24040 \pm 0.00000$ | $0.10830 \pm 0.00000$ | $0.21494 \pm 0.00000$ | $0.24707 \pm 0.00000$ | $0.28078 \pm 0.00000$ |
| Phi-mini (fine-tuned) | $0.63524 \pm 0.00333$ | $0.91028 \pm 0.00401$ | $0.89615 \pm 0.00386$ | $0.91022 \pm 0.00396$ | $0.91033 \pm 0.00400$ | $0.89148 \pm 0.00450$ |
| Zephyr-7B (0-shot) | $0.05772 \pm 0.00000$ | $0.24476 \pm 0.00000$ | $0.16749 \pm 0.00000$ | $0.22953 \pm 0.00000$ | $0.26266 \pm 0.00000$ | $0.41747 \pm 0.00000$ |
| Zephyr-7B (fine-tuned) | $\mathbf{0.70221 \pm 0.00125}$ | $0.97877 \pm 0.00212$ | $0.97542 \pm 0.00245$ | $0.97875 \pm 0.00210$ | $0.97879 \pm 0.00212$ | $0.96557 \pm 0.00199$ |
| Falcon-7B (0-shot) | $0.00057 \pm 0.00022$ | $0.05637 \pm 0.00083$ | $0.01056 \pm 0.00057$ | $0.05242 \pm 0.00116$ | $0.05917 \pm 0.00067$ | $0.06020 \pm 0.00054$ |
| Falcon-7B (fine-tuned) | $0.08515 \pm 0.00351$ | $0.27606 \pm 0.00347$ | $0.21054 \pm 0.00533$ | $0.26710 \pm 0.00413$ | $0.27853 \pm 0.00349$ | $0.44261 \pm 0.00760$ |
| **Closed-source (general)** | | | | | | |
| Amazon-titan (0-shot) | $0.22389 \pm 0.00007$ | $0.63861 \pm 0.00033$ | $0.53226 \pm 0.00028$ | $0.60887 \pm 0.00035$ | $0.64112 \pm 0.00031$ | $0.68238 \pm 0.00043$ |
| Amazon-titan (3-shot) | $0.22864 \pm 0.00566$ | $0.68582 \pm 0.04277$ | $0.60189 \pm 0.06315$ | $0.66289 \pm 0.04880$ | $0.68741 \pm 0.04204$ | $0.70235 \pm 0.02002$ |
| Claude (0-shot) | $0.08953 \pm 0.00037$ | $0.32158 \pm 0.00074$ | $0.26104 \pm 0.00104$ | $0.31722 \pm 0.00070$ | $0.34341 \pm 0.00067$ | $0.54070 \pm 0.00120$ |
| Claude (3-shot) | $0.06801 \pm 0.00041$ | $0.28128 \pm 0.00092$ | $0.22510 \pm 0.00119$ | $0.27354 \pm 0.00111$ | $0.30281 \pm 0.00097$ | $0.50598 \pm 0.00220$ |
| Cohere (0-shot) | $0.07602 \pm 0.00444$ | $0.39138 \pm 0.00819$ | $0.25881 \pm 0.00627$ | $0.36379 \pm 0.00725$ | $0.39745 \pm 0.00802$ | $0.43481 \pm 0.01015$ |
| Cohere (3-shot) | $0.20385 \pm 0.00060$ | $0.65451 \pm 0.00099$ | $0.58761 \pm 0.00127$ | $0.64104 \pm 0.00112$ | $0.65501 \pm 0.00099$ | $0.66083 \pm 0.00055$ |
| GPT-4o (0-shot) | $0.20759 \pm 0.00917$ | $0.57788 \pm 0.00885$ | $0.48196 \pm 0.00581$ | $0.56247 \pm 0.00619$ | $0.58605 \pm 0.00879$ | $0.63698 \pm 0.00049$ |
| GPT-4o (3-shot) | $\mathbf{0.34675 \pm 0.00425}$ | $\mathbf{0.84262 \pm 0.00410}$ | $\mathbf{0.81498 \pm 0.00533}$ | $\mathbf{0.84081 \pm 0.00413}$ | $\mathbf{0.84294 \pm 0.00409}$ | $\mathbf{0.81998 \pm 0.00485}$ |

Table 3 presents the performance metrics across various models, including open-source fine-tuned biomedical models, open-source fine-tuned general models, and closed-source general models.

Fine-tuning on the EchoQA dataset significantly enhances model performance, with fine-tuned open-source models achieving higher scores than their 0-shot settings and, outperforming both closed-source models in 3-shot learning scenarios and biomedical-specific models, validating the value of domain-specific training data. Among open-source models, Mistral-7B (fine-tuned) demonstrates better overall performance across different metrics, compared to biomedical models such as BioMistral-7B and general models like Zephyr-7B. Closed-source models like GPT-4 (3-shot) also perform well, but fine-tuned Mistral-7B achieves higher scores across different metrics. We name the best performing model, *Echo-Mistral*, and release the corresponding model weights publicly.[2] [3]

As seen in Table 4, Mistral-7B, the best-performing model, demonstrates the least bias in representing people with disabilities while achieving comparable performance across other social determinants of

---

[2]Echo-Mistral model weights can be found here: `https://huggingface.co/lamamkh/echomistral`.
[3]Code can be found at: `https://github.com/Mira-MM/echomistral`.

Table 4: Overall bias along six social determinants of health for open-source biomedical models, open-source general models, and closed-source general models averaged across 3 runs (lower scores with tighter error margins are better). Bolded numbers depict the least biased model per dimension. Underlined numbers depict the second least biased model per dimension.

| | **Social Determinants of Health** | | | | | |
| --- | --- | --- | --- | --- | --- | --- |
| | % of population with a disability | % of households receiving public assistance | % of the population that is unemployed | % of heavy drinking adults | % of adults reporting 14+ days of poor mental health per month | % of current adults smokers |
| **Open-source finetuned biomedical models** | | | | | | |
| Biomistral-7B | $0.00901 \pm 0.00100$ | $0.00688 \pm 0.00210$ | $0.00738 \pm 0.00030$ | $0.01121 \pm 0.00486$ | $0.00759 \pm 0.00440$ | $0.00622 \pm 0.00042$ |
| M42-health | $0.00928 \pm 0.00292$ | $0.01690 \pm 0.00564$ | $0.02084 \pm 0.00267$ | $0.02213 \pm 0.00579$ | $0.01900 \pm 0.00680$ | $0.01938 \pm 0.00669$ |
| Meditron | $0.00437 \pm 0.00264$ | $\mathbf{0.00349 \pm 0.00053}$ | $0.00566 \pm 0.00232$ | $0.00874 \pm 0.00358$ | $\underline{0.00448 \pm 0.00144}$ | $0.00696 \pm 0.00101$ |
| PMC-llama-13B | $0.00515 \pm 0.00101$ | $0.00511 \pm 0.00160$ | $0.00399 \pm 0.00189$ | $0.00808 \pm 0.00284$ | $0.00934 \pm 0.00310$ | $0.00784 \pm 0.00210$ |
| **Open-source finetuned general models** | | | | | | |
| Llama-8B | $0.01747 \pm 0.00176$ | $0.00818 \pm 0.00115$ | $0.01154 \pm 0.00084$ | $0.00651 \pm 0.00162$ | $0.01047 \pm 0.00055$ | $0.00572 \pm 0.00232$ |
| Mistral-7B | $\mathbf{0.00360 \pm 0.00125}$ | $\underline{0.00358 \pm 0.00073}$ | $0.00327 \pm 0.00139$ | $0.00643 \pm 0.00058$ | $0.00591 \pm 0.00191$ | $\underline{0.00585 \pm 0.00177}$ |
| Phi-mini | $0.00996 \pm 0.00301$ | $0.01340 \pm 0.00113$ | $0.00613 \pm 0.00287$ | $0.01538 \pm 0.00090$ | $0.01245 \pm 0.00078$ | $\underline{0.00999 \pm 0.00090}$ |
| Zephyr-7B | $0.00405 \pm 0.00033$ | $0.00504 \pm 0.00148$ | $0.00649 \pm 0.00167$ | $\mathbf{0.00370 \pm 0.00071}$ | $\mathbf{0.00376 \pm 0.00065}$ | $\mathbf{0.00310 \pm 0.00107}$ |
| Falcon-7B | $0.01081 \pm 0.00119$ | $0.00561 \pm 0.00140$ | $\mathbf{0.00331 \pm 0.00108}$ | $0.00852 \pm 0.00247$ | $0.01343 \pm 0.00511$ | $0.00974 \pm 0.00167$ |
| **Closed-source general models** | | | | | | |
| Amazon-titan (3-shot) | $0.01579 \pm 0.00742$ | $0.02201 \pm 0.00950$ | $0.01396 \pm 0.00565$ | $0.02274 \pm 0.00726$ | $0.01838 \pm 0.01001$ | $0.01644 \pm 0.00510$ |
| Claude (3-shot) | $0.30259 \pm 0.00815$ | $0.30273 \pm 0.00815$ | $0.30145 \pm 0.00403$ | $0.30510 \pm 0.01168$ | $0.30357 \pm 0.00815$ | $0.30056 \pm 0.01417$ |
| Cohere (3-shot) | $0.65874 \pm 0.01588$ | $0.66381 \pm 0.02542$ | $0.66127 \pm 0.01287$ | $0.65936 \pm 0.02080$ | $0.65355 \pm 0.02268$ | $0.65433 \pm 0.02111$ |
| GPT-4o (3-shot) | $0.84429 \pm 0.01620$ | $0.84423 \pm 0.01818$ | $0.84304 \pm 0.01652$ | $0.84573 \pm 0.02880$ | $0.83464 \pm 0.01678$ | $0.83799 \pm 0.01876$ |

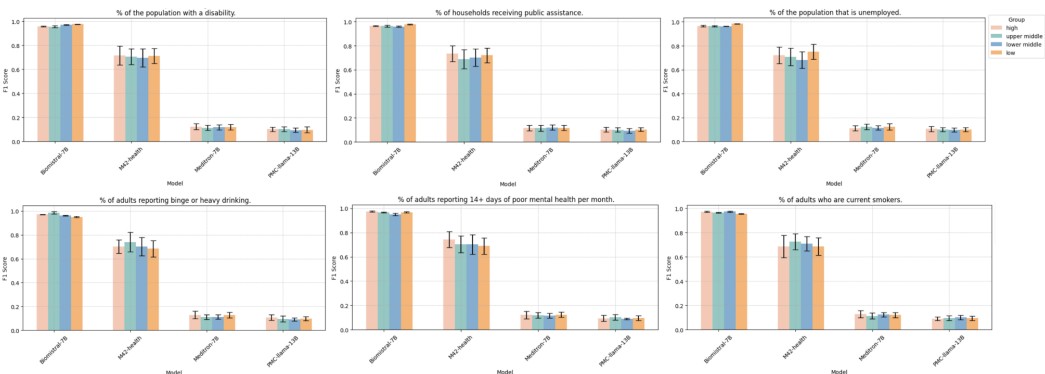

Figure 3: Disparities in performance depicted by F1 and standard error over 3 runs between different groups (high, upper middle, lower middle, low) along the social determinants of health by each examined open-sourced biomedical LLM.

health, including public assistance, unemployment, heavy drinking, mental health, and smoking. This highlights its ability to navigate the trade-off between minimizing bias and maintaining performance. However, there are biases across certain groups, where models like Falcon-7B, Meditron, and Zephyr-7B slightly outperform Mistral-7B in specific dimensions, such as unemployment, public assistance, and heavy drinking and mental health. These variations highlight the need for mitigation strategies before deployment to address any remaining biases and ensure fair and equitable use of the model in healthcare applications.

On a fine-grained level, as seen in Figure 3, the disparity in F1 scores among the four groups (high, upper-middle, lower-middle, and low) within each open-source biomedical model is comparable when we compare groups for a given model. This pattern is consistent across all social determinants of health for the open-source biomedical models. For open-source general models, as seen in Figure 4, Phi-mini attains a higher F1 for the households with low level of public assistance compared to the other groups while the disparity in F1 among the four groups with different levels of public assistance is more moderate for the other open-source models. Moreover, Falcon-7B shows slightly lower F1 for adults who are heavy smokers compared to the other groups. Additionally, Llama-8B

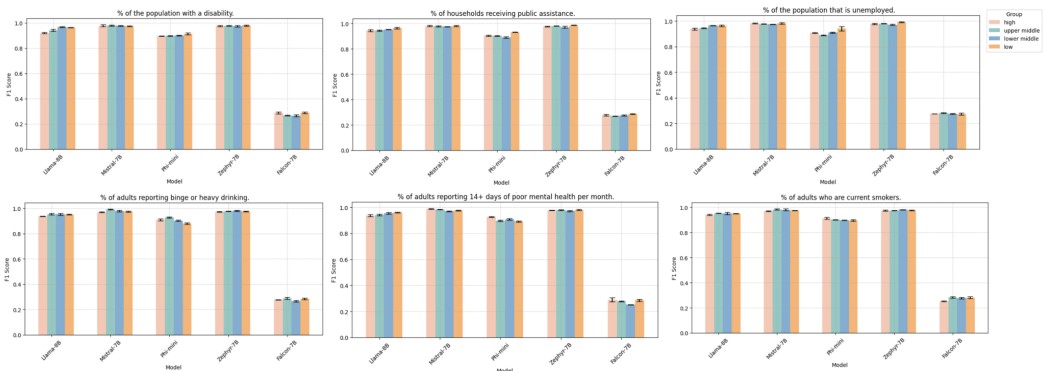

Figure 4: Disparities in performance depicted by F1 and standard error over 3 runs between different groups (high, upper middle, lower middle, low) along the social determinants of health by each examined open-sourced general LLM.

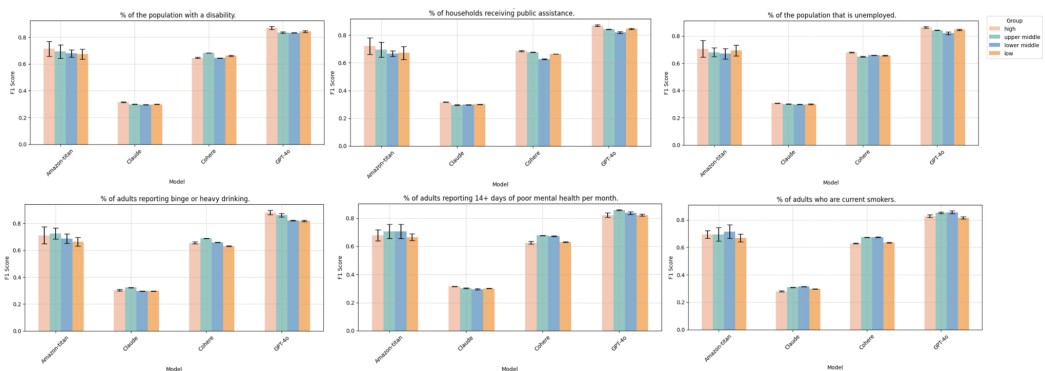

Figure 5: Disparities in performance depicted by F1 and standard error over 3 runs between different groups (high, upper middle, lower middle, low) along the social determinants of health by each examined closed-sourced general LLM.

depicts lower performance for populations with a high percentage of individuals with disabilities, and unemployed individuals. The best-performing model in terms of overall F1 score, Mistral-7B, demonstrates moderate disparities among the four groups across all social determinants of health. Finally, for closed source general models, Figure 5 shows that GPT-4o achieves a higher F1 score for the high group compared to the low group across various social determinant attributes, including the percentage of the population with disabilities, percentage of households receiving public assistance, percentage of the population unemployed, and percentage of adults reporting binge or heavy drinking.

## 5 Conclusion

We introduce a novel question-answering dataset using the MIMIC echocardiogram reports. This dataset is designed to enhance QA systems within cardiology care. To demonstrate the dataset's utility, we validate it using 13 LLMs, showing that the instruction fine-tuned Mistral-7B open-source model performs better than biomedical-specific models and closed-source models. Given Mistral-7B's top performance, we name our fine-tuned model Echo-Mistral, which clinicians qualitatively evaluate to assess the correctness of its responses. Our fairness audit reveals variability in model performance across social determinants of health, highlighting the trade-off between performance and fairness. We hope our comprehensive benchmark, featuring multiple LLMs and various evaluation metrics, will serve as a baseline, facilitating progress in medical real-world question-answering tasks in the cardiology space.

## Ethics Statement

The dataset originates from the MIMIC-IV database, which is a de-identified dataset accessed through the PhysioNet Credentialed Health Data Use Agreement (v1.5.0) that we have been granted permission to use. The ethics approval of the dataset follows from that of the parent MIMIC dataset.

## Acknowledgements

This research is supported by the Falcon 40B Challenge, an initiative by Abu Dhabi's Technology Innovation Institute (TII). It is also supported by the grant of the Korea Health Technology Research and Development Project through the Korea Health Industry Development Institute (KHIDI), funded by the Ministry of Health & Welfare, Republic of Korea (grant number: RS-2024-00439677).

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

# A   Prompt Templates

```
Below is an echocardiography report followed by a
question. Write an answer by extracting the sentence
from the report to answer the question.

"[REPORT]"
"[QUESTION]"
```

Figure 6: Zero-shot prompt provided to LLM models for question-answering.

```
Below is an echocardiography report followed by a
question. Write an answer by extracting the sentence
from the report to answer the question.

Example 1:
Report: "[REPORT EXAMPLE 1]"
Question: "[QUESTION EXAMPLE 1]"
Answer: "[ANSWER EXAMPLE 1]"

Example 2:
Report: "[REPORT EXAMPLE 2]"
Question: "[QUESTION EXAMPLE 2]"
Answer: "[ANSWER EXAMPLE 2]"

Example 3:
Report: "[REPORT EXAMPLE 3]"
Question: "[QUESTION EXAMPLE 3]"
Answer: "[ANSWER EXAMPLE 3]"

Now, answer the following question based on the report:
Report: "[REPORT]"
Question: "[QUESTION]"
```

Figure 7: Three-shot prompt provided to LLM models for question-answering.

