# OpenReview forum: "EchoQA: A Large Collection of Instruction Tuning Data for Echocardiogram Reports"
_NeurIPS.cc/2024/Workshop/SafeGenAi — SafeGenAi Poster_

### Official Review · Reviewer_sB5x · 2024-10-09
**This paper introduces a large dataset aimed at improving QA systems in cardiology. They also produce extensive evaluations of the performance of various LLMs on cardiological QA, both after fine-tuning on their data or being frozen. However, the paper is not particularly relevant to the AI Safety.**

**Rating:** 5
**Confidence:** 3

**Review:**

This paper presents an extensive dataset consisting of 765605 question-answer pairs, which was carefully designed with clinicians. They also use 12 LLMs (biomedically specific, general, frozen and fine-tuned), to produce an analysis of their perfomance across natural-language generation metrics, and fairness. Although the paper is presented clearly and is a useful contribution to the medical AI community it is not particularly relevant to the present workshop.

Pros:
- The dataset is high-quality, extensive, diverse and was curated with specialist clinicians.
- The authors were thorough in the set of experiments they conducted, and consequently produced some interesting insights (e.g. the superior performance of Mistral-7B above other models.)
- The paper is very clear which means the experiments can be reproduced. And putting the data on PhysioNet will make it easily accessible.

Cons:
- Relevance to Workshop Theme: While the paper introduces a valuable dataset and conducts thorough experiments, its primary focus is not on safety concerns central to the workshop. The fairness evaluation, although present, is not the central thesis, which means that although it is a useful contribution to medical AI, it is slightly less for Safe AI.

- Emphasis on Performance Metrics: The paper's focus on natural language generation metrics is informative for the medical AI community but doesn't directly contribute to the discourse on safety in generative AI. A deeper exploration of bias assessment and mitigation strategies would have made it more pertinent to the workshop's objectives.

---

### Official Review · Reviewer_iv2t · 2024-10-09
**New benchmark fills important niche in LLMs for healthcare applications. Proof-of-concept provided; fairness audit design is exemplary.**

**Rating:** 8
**Confidence:** 4

**Review:**

**Summary**

The paper introduces EchoQA, a question-answering dataset for instruction-tuning and zero-shot LLM evaluation for echocardiogram reports, a common task in cardiology. Performance of multiple generalist and biomedical-domain LLMs are evaluated in an instruction-tuning setup, and zero-shot evaluation is conducted on commercial LLMs, providing proof-of-concept for the usability of the benchmark. Fairness auditing with respect to social determinants of health is also conducted for transparency.

**Strengths**
* Despite the plethora of new benchmarks for LLMs, the paper makes a good case for why this benchmark fills an unmet need (namely cardiology tasks + LLMs)
* The paper validates the usability of the benchmark by applying it to instruction-tuning and zero-shot evaluations — good proof of concept. Table 2  has some pretty stunning results comparing zero-shot w/ fine-tuning, highlighting serious gaps in current "out-of-the-box" models.
* The fairness audit is an excellent touch and very necessary in applications of GenAI to healthcare. In particular, rather than simply slicing by standard "protected attributes," which is a rather Swiss-army-knife approach to fairness eval, the authors *specifically choose to analyze based on slices that are directly motivated by the underlying application* (i.e., social determinants of health). This is an excellent example of a fairness audit for ML + healthcare.
* Release via PhysioNet is a great way to scale the impact of such a benchmark.

**Weaknesses**
* I'm a little confused by the question-generation process. Specifically, which questions are generated by experts (L147-149), and which ones are extracted via the text analysis (L164-165)? Are clinicians simply guiding the question extraction, or writing some questions in the dataset themselves?
* The related works could benefit from discussing modeling and benchmarking in LLMs + cardiology (albeit not echocardiograms), such as:
	* Models: [EchoGPT](https://www.medrxiv.org/content/10.1101/2024.01.18.24301503v3), [HEART](https://openreview.net/forum?id=fGVQgxvrzI)
	* Benchmarking: [Low Perf. of ChatGPT on Echo Boards](https://www.jacc.org/doi/abs/10.1016/j.jcmg.2023.09.004) [Review of LLMs+Cardiology](https://www.medrxiv.org/content/10.1101/2024.09.01.24312887.abstract)
* (minor) The text in Fig. 2 in extremely small. Consider decreasing figsize.